# Risk factors and true prevalence of bovine tuberculosis in Bangladesh

**Md. Nazimul Islam**[1,2], **Mohammad Kamruzzaman Khan**[1,3], **Mohammad Ferdousur Rahman Khan**[4], **Polychronis Kostoulas**[5], **A. K. M. Anisur Rahman**[1], **Md. Mahbub Alam**[1]*

**1** Faculty of Veterinary Science, Department of Medicine, Bangladesh Agricultural University, Mymensingh, Bangladesh, **2** Department of Livestock Services, Dhaka, Bangladesh, **3** Department of Community Medicine, Mymensingh Medical College, Mymensingh, Bangladesh, **4** Faculty of Veterinary Science, Department of Microbiology and Hygiene, Bangladesh Agricultural University, Mymensingh, Bangladesh, **5** Faculty of Public Health and One Health, Laboratory of Epidemiology & Artificial Intelligence, School of Health Sciences, University of Thessaly, Volos, Greece

* asamahbub2003@yahoo.com

**Data Availability Statement:** All relevant data are within the manuscript and its Supporting Information files.

## Abstract

Bovine tuberculosis (bTb) is endemic in Bangladesh but the true prevalence has not yet been reported. Our objectives for this study were to determine the true prevalence and identify risk factors for bTb at the animal- and herd-level in Bangladesh. A total of 510 cows were randomly selected during January 2018 to December 2018. Caudal fold (CFT) and comparative cervical tuberculin tests (CCT) were serially interpreted. Animal- and herd-level risk factor data were collected using a pre-tested questionnaire. The hierarchical true prevalence of bTb was estimated within a Bayesian framework. The herd- and animal-level risk factors were identified using mixed effects logistic regression. The apparent prevalence of bTb was 20.6% [95% Confidence Interval (CI): 17.3; 24.3] based on CFT. The animal-level true prevalence of bTb was 21.9 (13.0; 32.4). The herd-level true prevalence in different regions varied from 41.9% to 88.8%. The region-level true prevalence was 49.9 (13.8; 91.2). There is a 100% certainty that herds from Bhaluka and Mymensingh Sadar upazilas are not free from bTb. The odds of bTb were 3.9 times (1.2; 12.6) higher in herds having more than four cows than those with ≤ 4 cows. On the other hand, the risk of bTb was 3.3 times higher (1.0; 10.5) in non-grazing cows than grazing cows. Crossbred cows were 2.9 times (1.5; 5.9) more likely to be infected with bTb than indigenous cows. The risk of bTb in animals with cough was 2.3 times (1.2; 4.3) higher than those without cough. Crossbred, non-grazing cows with cough should be targeted for bTb surveillance. Herds of the Mymensingh, Sadar and Bhaluka regions should be emphasized for bTb control programs. Estimation of Bayesian hierarchical true prevalence facilitates identification of areas with higher prevalence and can be used to indicate regions that where true prevalence exceeds a pre-specified critical threshold.

## Introduction

Bovine tuberculosis (bTb) is a contagious, notifiable, granulomatous, chronic bacterial disease [1,2]. It is a zoonotic disease and the disease is still widely distributed and often neglected in most

**Funding:** This work was funded by the United States Department of Agriculture. Grant number: USDA 2001 Section 41 (b). The corresponding author (MMA) received this grant. Two Ph.D. students [the first (MNI) and second (MKK) authors] received monthly scholarship for three years from this grant. The funders had no role in study design, data collection and analysis, decision to publish, or preparation of the manuscript.

**Competing interests:** The authors have declared that no competing interests exist.

developing countries [3]. Bovine TB is caused mainly by *Mycobacterium bovis*, a member of the *Mycobacterium tuberculosis* complex (MTC) which consists of *M. tuberculosis*, *M. bovis*, *M. bovis* BCG, *M. africanum*, *M. caprae*, *M. microti*, *M. pinnipedii*, *M. leprae*, *M. canetti*, *M. orygis*, *M. mungi* and *M. suricattae* [4–7]. However, only *M. orgyis* has been reported so far from cattle in one dairy farm in Bangladesh [8]. The clinical signs of bTb are not specifically distinctive and can include weakness, anorexia, emaciation, dyspnea, hacking cough and enlargement of lymph nodes particularly with advanced stage of the disease [1]. Evaluations of economic losses due to bTb is not yet addressed systematically on livestock production [9]. The economic losses of bTb was due to 10 to 25% loss of productive efficiency, 10–18% decreased milk yield,15% reduction in meat production, increased edible organs condemnation and increased mortality [10,11].

Two thirds of the global total TB burden is in eight countries: India (27%), China (9%), Indonesia (8%), the Philippines (6%), Pakistan (6%), Nigeria (4%), Bangladesh (4%) and South Africa (3%) [12].The global prevalence of human TB caused by *M. bovis* was estimated to be 3.1% of all human TB cases worldwide [13]. The prevalence of human TB due to *M. bovis* was 0.5–2% in developed countries and 10–15% in developing countries [14–16].

Several studies reported the prevalence of bovine tuberculosis by tuberculin test (TST) from different geographic locations in Bangladesh. A 5.9% TST positive cattle in Pabna district [17] and 3.05% TST positive cattle in Mymensingh district [18]. The prevalence of bTb in breeding bulls at central cattle breeding and dairy farm and Bangladesh Livestock Research Institute farm, Savar, Dhaka were reported to be 27.5%, [19]; 7.1%, [20] respectively. TST positive cattle reported at private and government dairy farms in different districts of Bangladesh was 4.08% [21]. In Bangladesh the prevalence of bTb based on ELISA and rapid test were 5.9% [22], 5.9% [23], 7.8% [24], 30% [25]. The prevalence of bTb in cattle in India and Pakistan were reported be 7.3% and 6.4%, respectively [26,27].

The risk factors for bTb reported so far were herd size, poor husbandry and sanitary practices, the introduction of a new animal from an unknown source, breed, age group, animals having close contact with other animals, communal grazing and watering [27,28].

Diagnostic tests usually used for bTb like caudal fold tuberculin (CFT) test and comparative cervical tuberculin are not perfect and hence reported prevalence estimates are, in reality, apparent prevalence [29].

None of the previously conducted studies reported the true prevalence of bTb in Bangladesh, Thus, our objectives for this study were to determine the true prevalence of bTb and to identify the animal and herd level risk factors for bTb in Mymensingh district of Bangladesh.

## Materials and methods

### Ethics statement

The study protocol was approved by the Animal Welfare and Experimentation Ethical Committee (AWEEC) of Bangladesh Agricultural University (AWEEC/BAU/2017/08). Oral consent was taken from the owner/manager of the cattle farm before undertaking tuberculin skin test and data collection as well.

### Study area, animal and study design

The study was conducted in Mymensingh district located in between 24˚15' and 25˚12' north latitudes and in between 90˚04' and 90˚49' east longitudes. Mymensingh is an ancient district consisting of thirteen sub-districts/upazilas, 146 unions/sub-upazila, and 2700 villages. Ten out of the thirteen upazilas were randomly selected for this cross-sectional study that was conducted from January 2018 to December 2018. The study included both semi-intensive and extensive dairy farms with Holstein Friesian, Jersey, Sahiwal crossbred and Indigenous cows.

In this study seventeen union/municipality, fifty-nine villages, eleven larger and one hundred seventy-eight smaller herds were included.

## Sampling and sample size

At least one union from each upazila and at least three villages from each selected union were selected randomly for this study. In the case of a smaller herd at least two bTb suspected cows were tested where at least ten bTb suspected cows were tested for larger herds. Calves less than 6 months, advanced pregnant, weak and extremely emaciated animals were excluded from the study.

Sample size was calculated based on the following formula given in Eq (1).

$$n = \frac{1.96^2 P(1 - P)}{d^2} \tag{1}$$

Where $P$(expected prevalence) = 0.10, $d$(precision) = 0.028. These assumptions produce a sample size of 441. Finally a total of 510 cattle were tested.

## Tuberculin test

The caudal fold tuberculin (CFT) test and comparative cervical tuberculin (CCT) test have been conducted in bTb suspected cattle for the screening of bovine tuberculosis in the study area. A herd has been considered positive if it has at least one positive tuberculin reactor. The same person has conducted the entire process of tuberculin testing and reading of the result to avoid bias related to injection and reading technique.

**Caudal fold tuberculin (CFT) test.** The primary screening was done with the CFT test to identify cattle infected with bovine TB. A 0.1 ml (Containing 2,500 IU/0.1ml purified protein derivative from the culture of *Mycobacterium bovis*, strain AN-5, CZ Veterinaria, Pontevedra, Spain) bovine PPD administered at intradermal into the skin of the caudal fold (the fold of skin at the base of the tail) using 1 ml disposable (10 graduations) syringe (GETWELL, BD). Before administration, the injection site was cleaned and disinfected properly with Hexisol (ACI, BD) and measured the thickness with Vernier caliper and recorded. The response to the CFT injection was done by inspecting and palpating the injection site 72 hours later. The difference of thickness > 4mm was considered a positive reactor [1,30]

**Comparative cervical tuberculin (CCT) test for bTb.** Only CFT positive reactor was considered for the CCT test and the CCT injection administered within the 10 days or more than 60 days following the CFT. Two sites on the left side of the mid-neck, 12 cm apart were shaved (2" X2") and disinfected properly with Hexisol (ACI, BD). The skin thickness was measured with a Vernier caliper and recorded. The upper site was injected with 0.2 ml avian PPD (Containing 25,000 IU/ml purified protein derivative of *Mycobacterium avium*, strain D4 ER CZ Veterinaria, Pontevedra, Spain) using 1 ml disposable (10 graduations) syringe (GETWELL, BD) and 0.1 ml bovine PPD administered at the lower site at the route of intradermal. After 72 hours, the skin fold thicknesses at both injection sites were re-measured and difference of thickness was recorded. A skin reaction was considered positive when the skin thickness increase at the bovine PPD site of injection was > 4mm than the reaction at the avian PPD injection site [1,30].

## Data collection

Structured questionnaires were developed, pre-tested, and finalized for the data collection. Pre-testing was done by an interview in which all aspects of the questionnaire were discussed. Cattle owners or attendants were interviewed according to their willingness. The effects of

different management practices in the herd and animal level were assessed through direct interviewing and after verbal consent on the first day of testing for bovine TB. Different husbandry practices were explored. Questionnaires included closed and open questions on animal level (age, breed, sex, BCS, pregnancy, parity, lactation stage, vaccination, deworming, coughing, diseases status) and herd level (livestock husbandry and household characteristics, herd size and structure, presence of other livestock, mixing of cattle and other livestock at night, grazing system, reproduction, purchasing of animals, use of disinfection, visitors entrance, disposal of cow dung, presence of stagnant water, presence of wild life, cleanliness of periphery, presence of farm) data.

## Data analysis

**True prevalence.**   The hierarchical true prevalence of bovine tuberculosis was estimated, within a Bayesian framework, based on the method previously described [31]. Briefly, the model allows for the estimation of the true prevalence of bTb, adjusting for the accuracy of the diagnostic process. The model estimates the animal-level prevalence within infected herds, the herd-level prevalence within infected regions and the region-level prevalence. Further, the probability that each herd and/or region is free from bTb is calculated, as well as the probability that the true prevalence of bTb is not exceeding a pre-specified critical threshold. Model description and the OpenBUGS code, with step by step explanations, is attached as S1 File. The data used for the estimation of hierarchical true prevalence is provided as S2 File.

**Prior information for prevalence, sensitivity and specificity.**   Prior specification was based on available information which was relevant to the target population and independent from the current study. Beta distributions were used as priors for animal-level prevalence, the sensitivity and specificity of CFT-CCT. As there was no previous report on the herd- and region-level bTb prevalences, we used uniform Beta priors [Beta(1,1)] priors for these prevalences. Beta distributions for the priors on animal level prevalences, sensitivity and specificity of CFT-CCT were calculated using the 'findbeta' and 'findbetamupsi' functions of the package 'PriorGen' [32] in R 4.0.2. [33]. The mean prevalence of a bTb for the herds within an area/region is reported to be 0.095 and we are 95% confident that it is not more than 0.18. Within this area, we are also confident that 90% of all herds have a prevalence less or equal to 0.35 and we are 95% certain that it does not exceed 0.45 [17–19,21]. The mean values of the reported sensitivity and specificity of caudal fold test (CFT) were 80% and 90% and we can be 95% sure that they are higher than 51% and 80%, respectively [34–39]. Whereas the mean values of the reported sensitivity and specificity of CFT-comparative cervical tuberculin tests (CFT-CCT) were 53% and 97% and we can be 95% certain that they are higher than 46% and 94%, respectively [34–39].

**Sensitivity analysis.**   The influence of prior information on the estimates of the model parameters was assessed considering three additional, different sets of prior information: (a) uniform priors in the range 0–1 for Ses and Sps (i.e. $\beta(1,1)$ distribution); (b) uniform priors in the range 0–1 for Sps and the informative priors for Ses used for the primary analysis and (c) uniform priors in the range 0–1 for Ses and the informative priors for Sps used for the primary analysis [40].

**Mixed-effect logistic regression analyses.**   Two separate manual forward mixed-effect multiple logistic regression models were used to identify risk factors for bTb at animal and herd levels. If any cow in a herd was positive with CFT then that cattle (and herd) were defined as positive for bovine tuberculosis. All continuous variables (herd size, age of the animal and body weight) were categorized prior to logistic regression analysis. Initially, univariable mixed-effect logistic regression analyses were performed by including herd and upazila as random intercepts for animal and herd level, respectively in R package "lme4" [41]. The bTb status was used as the response and each risk indicator variable in turn as an explanatory variable

in the model. Any explanatory variable associated with bTb status with a p-value of ≤0.20 was selected for multiple mixed effect logistic regression analysis. Co-linearity among explanatory variables was assessed by calculating a Cramer's phi-prime statistic (R package "VCD," "associates" function). A pair of variables were considered collinear if Cramer's phi-prime statistic was >0.70 [42]. The best univariable model was selected based on the lowest Akaike's information criterion (AIC) value. Then the remaining variables were added in turn, based on AIC. The final model selected had the lowest AIC. Confounding was checked by observing the change in the estimated coefficients of the variables that remained in the final model by adding a non-selected variable to the model. If the inclusion of this non-significant variable led to a change of more than 25% of any parameter estimate, that variable was considered to be a confounder and retained in the model [43]. The two-way interactions of all variables remaining in the final model were assessed for significance based on AIC values [43]. The intraclass correlation coefficient (ICC), which is a measure of the degree of clustering of bTb positive cattle belonging to the same herd/district, was estimated using the formula: $ICC_{Herd/Upazila} =$

$\partial^2_{Herd/Upazila} / \left( \partial^2_{Herd/Upazila} + {\pi^2}/{3} \right)$ [44]. The 95% confidence interval of the ICC was bootstrapped using the "bootMer" function of the R package "lme4" [41]. The areas under the Receiver Operating Characteristic (ROC) Curves were constructed for both herd level and animal level models to determine their predictive ability.

The Bayesian hierarchical model was run in OpenBUGS [45] with a burn-in period of 50 000 iterations and estimates were based on a further 50 000 iterations using three chains. The convergence of the model was assessed by time-series plots, Gelman Rubin convergence diagnostics, autocorrelation plots and Monte Carlo standard errors [46].The mixed effect logistic regression analyses were performed in R 4.0.2 [33]. The data used for the identification of herd and animal level risk factors are provided as S3 and S4 Files, respectively.

## Results

### Descriptive results

The CFT was performed in 510 cow from 189 herds located in 10 upazilas of Mymensingh district. CCT was conducted in CFT positive animals. The median (interquartile range, IQR) herd size was 4 (4–5). The demographic characteristics of the farmers and farm management practices are presented in Table 1. About 97.4% famers were male and 25% farmers did not receive any formal education. Nearly 98% herds are open to visitors. Only 16.4% and 11.1% farmers reported the history of tuberculosis in their herds and family, respectively.

The median (IQR) age and body weight of the sampled cows were 5 (3.3–8) years and 250 (200–300) Kg, respectively. Fifty-nine percent animals were vaccinated against Foot-and-Mouth disease and 78% were dewormed routinely.

### Prevalence of bovine tuberculosis

The apparent prevalence of bTb were 20.6% (95% CI: 17.3; 24.3) and 7.3 (5.2; 9.9) based on CFT and CFT-CCT, respectively. Out of 105 CFT positive animals, 35% were CCT positive (Table 2). Twenty out of 189 herds (10.6%) from four upazilas (Bhaluka:12, Mymensingh Sadar: 6, Muktagacha: 1 and Trishal:1) were CCT positive.

### True prevalence of bovine tuberculosis

The mean true prevalence of bTb within any infected herd in Mymensingh district was 21.9% [95% credibility intervals (CrIs.): 13.0; 32.4]. The probability of the true herd-level prevalence

**Table 1. Demographic characteristics of the farm owner and the farm management practices.**

| Variables | Category | Number (%) |
|---|---|---|
| Gender | | |
| | Male | 184 (97.4) |
| | Female | 5 (2.6) |
| Education | | |
| | No formal education | 47 (24.9) |
| | Primary | 72 (38.1) |
| | Higher secondary | 65 (34.4) |
| | University | 5 (2.6) |
| Location of cowshed | | |
| | Far away from house hold | 14 (7.4) |
| | In the same premises | 175 (92.6) |
| Introduction of new animals in the herd | | |
| | No | 120 (63.5) |
| | Yes | 69 (36.5) |
| Cowshed floor sanitary status | | |
| | Poor | 165(87.3) |
| | | 24 (12.7) |
| Feed | | |
| | Straw | 189 (100) |
| | Green | 174 (92.1) |
| | Rice and wheat bran | 93 (49.2) |
| | Rice, wheat bran and molasses, vitamin mineral premixes | 23 (12.2) |
| Biosecurity status | | |
| | Poor | 185 (97.9) |
| | Good | 4 (2.1) |
| Cow dung disposal | | |
| | Near farm | 178 (94.2) |
| | Biogas | 11 (5.8) |
| Ventilation status | | |
| | Defective | 178 (94.2) |
| | Good | 11 (5.8) |
| History of tuberculosis in family | | |
| | Yes | 21 (11.1) |
| | No | 168 (88.9) |
| History of tuberculosis in farm | | |
| | Yes | 31 (16.4) |
| | No | 158 (83.6) |

being zero were estimated to be 0, 0.712, 0.696, 0.714, 0.766, 0.455, 0.729, 0, 0.694 and 0.247 for Bhaluka, Dhobaura, Fulbaria, Haluaghat, Ishwarganj, Muktagacha, Phulpur, Mymensingh Sadar, Tarakanda and Trishal, respectively. Further, the probability that herd-level prevalence is higher than a critical threshold, for example 50%, was also calculated. Here, the probability that the herd-level prevalence is higher than 50% is 0.002, 0.943, 0.936, 0.942, 0.975, 0.861, 0.958, 0.271,0.933, 0.993 and 0.596 for Bhaluka, Dhobaura, Fulbaria, Haluaghat, Ishwarganj, Muktagacha, Phulpur, Mymensingh Sadar, Tarakanda and Trishal, respectively. Obviously, we can be certain, with 95% confidence, that Bhaluka and Mymensingh Sadar have a herd-level

**Table 2. Apparent prevalence of bovine tuberculosis in Mymensingh district based of caudal fold and comparative cervical tuberculin tests.**

| Tuberculin test | Positive/Tested | Prevalence (95% Confidence Interval) |
|---|---|---|
| Caudal fold | 105/510 | 20.6 (17.3; 24.3) |
| Comparative cervical | 37/510 (serial) | 7.3 (5.2; 9.9) |

prevalence less than 50%. The probability that regions are not free from bTB was presented in Table 3. Finally, the mean regional prevalence of bTb was 49.9 (13.8; 91.2) (Table 3).

## Results of sensitivity analyses

The estimates under alternative prior specification (sensitivity analyses) are presented in Table 4. The influence of alternative priors on mean true prevalence of bTb within any infected herd and mean regional prevalence of bTb were evaluated. The mean true prevalence of bTb within a herd and the mean regional prevalence of bTb were similar in different alternative priors.

## Herd and animal level risk factors

In univariable analyses, herd size, presence of other animals in the herd, management system, grazing and use of mosquito net were significant (P≤0.20) at herd level (S5 File). Similarly, at animal level only breed and coughing were significant (P≤0.20) in univariable analyses (S6 File).

**Table 3. Animal-, herd- and region-level apparent and true prevalence of bovine tuberculosis in Mymensingh district, Bangladesh.**

| Level | Mean apparent prevalence (95% Confidence Interval) | Mean true prevalence (95% Credibility Interval) |
|---|---|---|
| Animal-level | 7.3 (5.2; 9.9) | 21.9 (13.0; 32.4) |
| **Herds in regions** | | |
| Bhaluka | 12/37 = 32.4 (18.6; 49.9) | 89.2 (65.1; 99.7) |
| Dhobaura | - | 44.1 (1.7;96.7) |
| Fulbaria | - | 44.1 (1.6;96.6) |
| Haluaghat | - | 44.2 (1.7;96.7) |
| Ishwarganj | - | 43.5 (1.4; 96.8) |
| Muktagacha | 1/21 = 4.8 (0.2; 25.9) | 41.9 (2.5; 95.6) |
| Phulpur | - | 43.5 (1.6; 96.7) |
| MymensinghSadar | 6/37 = 16.2 (6.8; 32.7) | 63.8 (23.1; 97.8) |
| Tarakanda | - | 44.2 (1.6; 96.7) |
| Trishal | 1/11 = 9.1 (4.8; 42.9) | 52.2 (4.7; 97.3) |
| **Regions in district** | 4/10 = 40.0 (13.7; 72.6) | 49.9 (13.8; 91.2) |
| Probability that regions are not free from bTb | Mean | |
| Bhaluka | 100 | |
| Dhobaura | 28.8 | |
| Fulbaria | 30.4 | |
| Haluaghat | 28.6 | |
| Ishwarganj | 23.4 | |
| Muktagacha | 54.5 | |
| Phulpur | 27.1 | |
| MymensinghSadar | 99.9 | |
| Tarakanda | 30.6 | |
| Trishal | 75.3 | |

**Table 4. True prevalence estimates under alternative prior specifications (sensitivity analysis).**

| Models and Tests | Prevalence (95% Credibility Interval |
|---|---|
| Uniform priors in the range of 0 to 1 for Ses and Sps (i.e. β(1,1) distribution) | |
| Animal in herds | 18.4 (10.3; 29.6) |
| Regions in district | 46.7 (12.1; 88.4) |
| Uniform priors in the range of 0 to 1 for Sps and the informative priors for Ses used for the primary analysis | |
| Animal in herds | 21.2 (10.3; 32.3) |
| Regions in district | 48.9 (13.1; 90.3) |
| Uniform priors in the range of 0 to 1 for Ses and the informative priors for Sps used for the primary analysis | |
| Animal in herds | 17.9 (9.2; 29.1) |
| Regions in district | 47.0 (12.7; 88.2) |

Se: Sensitivity; Sp: Specificity.

Finally, herd size and grazing as herd level (Table 5) and breed and coughing (Table 6) as animal level were identified as risk factors for bTb. The odds of bTb were 3.9 times (95% CI: 1.2; 12.6) higher in herds having more than four cows than those with ≤ 4 cows. On the other hand, the risk of bTb was 3.3 times higher (95% CI: 1.0; 10.5) in non-grazing cows than grazing cows. Crossbred cows were 2.9 times (95% CI: 1.5; 5.9) more likely to be infected with bTb than indigenous cows. The risk of bTb in animals with cough was 2.3 times (95% CI: 1.2; 4.3) higher than those without coughing.

No confounding variable was found. All two-way interaction of the variables retained in the final mixed-effect models were non-significant.

The area under ROC curves for herd- and animal-level final models were 90.6% and 91.0%, respectively which indicated that models fitted the data well and had a high predictive ability to discriminate test-positive and test-negative herds and animals (Fig 1A and 1B). The intra-class correlation coefficient (ICC) for herd and upazila level were 0.47 and 0.43, respectively.

## Discussion

Two risk factors at the animal- and two at the herd-level were identified. The true prevalence of bTb at the animal-, herd- and region-level was also estimated for the first time in Bangladesh. The results suggest that crossbred cows with cough in larger herds should be screened for bTb. Control measures like test and segregation/slaughter with compensation especially in herds with higher animal level prevalence will help reduce the burden of bTb in cattle and thereby reduce the risk of zoonotic transmission.

**Table 5. Herd-level risk factors retained in the final multivariable mixed effect logistic regression model for bovine tuberculosis in Mymensingh district, Bangladesh.**

| Variable | Category | Estimate | SE | Odds ratio (95% Confidence Interval) | P-value |
|---|---|---|---|---|---|
| Herd size | | | | | |
| | ≤ 4 | - | - | 1 | |
| | > 4 | 1.37 | 0.59 | 3.9 (1.2; 12.6) | 0.02 |
| Grazing | | | | | |
| | No | 1.18 | 0.59 | 3.3 (1.0; 10.5) | 0.04 |
| | Yes | - | - | 1 | |

Variance random effect (Region/Upazila) = 3.01; Intraclass Correlation Coefficient = 0.47.

**Table 6. Animal-level risk factors retained in the final multivariable mixed effect logistic regression model for bovine tuberculosis in Mymensingh district, Bangladesh.**

| Variable | Category | Estimate | SE | Odds ratio (95% Confidence Interval) | P-value |
|---|---|---|---|---|---|
| Breed | | | | | |
| | Indigenous | - | - | 1 | |
| | Crossbred | 1.09 | 0.34 | 2.9 (1.5; 5.9) | 0.001 |
| Coughing | | | | | |
| | No | - | - | 1 | |
| | Yes | 0.82 | 0.32 | 2.3 (1.2; 4.3) | 0.01 |

Variance random effect (Herd) = 2.5, Intraclass Correlation Coefficient = 0.43.

The apparent prevalence of bTb based on CFT (20.6%) were about three times higher than that based on CCT (7.3%). Only CFT positive cattle were further confirmed by CCT. An increase in skin thickness ≥4mm after 72 hrs is considered as positive in CFT where the same increase in skin thickness in bovine site than avian site is considered as positive in CCT. This difference in test interpretation may be responsible for lower prevalence in CCT than CFT.

True prevalence is an essential piece of information for the design of disease control strategies. The true prevalence of bTb at the animal-level was estimated to be 21.9% (13.0; 32.4). Our estimates are comparable to previous estimates but have the advantages that they adjust for the imperfect sensitivity and specificity of the diagnostic process [17–19,21]. In a previous study, the overall true prevalence of bTB was reported to be 11.8 (2.1–20.3%) which is also comparable to our result as the 95% credible intervals overlap [47]. Skin test results are often associated with the sensitization of the animals to *M. avium* subsp *paratuberculosis* and atypical mycobacteria [48]. However, initially we screened all selected animals by CFT [with bovine PPD] then CFT positive animals were further confirmed by comparative cervical tuberculin test [with both avian and bovine PPD] which mostly excluded responses from *M. avium* subsp *paratuberculosis*. Moreover, we used serial interpretation of the diagnostic tests which decreases false positive results and hence increases the positive predictive value of the diagnostic tests [40]. Indeed, the sensitivity of this test should be evaluated in future control programs by isolating *M. bovis* from the post-slaughter material of test positive animals.

The herd-level prevalence of bTb was higher in Bhaluka (89.2%) and Mymensingh Sadar (63.8%). We are also 100% certain that herds of these two regions are not free from bTb. Test positive animals in the herds of these upazilas should be segregated and slaughtered upon compensation.

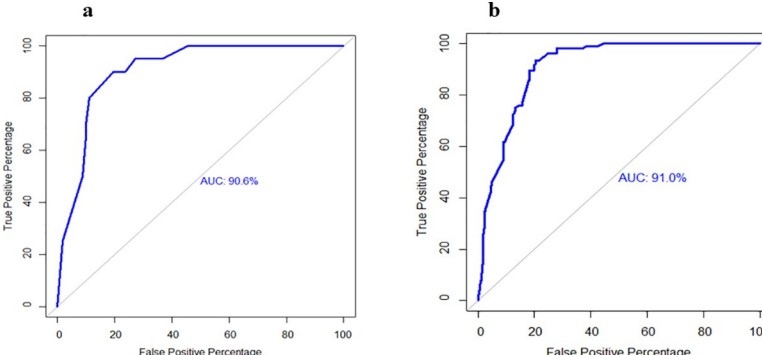

**Fig 1.** Receiver operating characteristic curve of the final multivariable mixed effect logistic regression models, a: at herd level, b: at animal level.

The herd-level bTb prevalence was higher in larger herds (>4 animals) than smaller herds. Lack of adequate floor space, improper floor sanitation, and faulty ventilation system or confinement of the animals within the shed, inadequate management of farms may be the principle variables to increase the susceptibility of bTb in larger herds in the current study. Several reports supported this hypothesis that bTb is more prevalent in larger herds than smaller ones [23,27,47,49–56].

We observed that about 35% of farmers did not graze their animals in the field and kept the animals within the shed. It was also found that about 94% of sheds had defective ventilation systems in the study area which might increase the risk of bTb transmission [57,58]. Close contact with other animals and keeping animals within the shed are known to be important risk factors for bTb. This was also supported by several reports [59–61].

To increase national milk production, artificial insemination has been used with high-productivity breeds–especially Holstein Friesian–semen in Bangladesh since the 1980s [62] and the number of crossbred cattle population is gradually increasing and now around 15% [63]. In the present study, about 78.6% were crossbred cow. The susceptibility to bTB has been reported to vary based on genetics of the host like higher susceptibility to TB among *Bos taurus* which are more productive than *Bos indicus* [64–68]. Additionally, bTB susceptibility has been found to be associated with a gene called TauT [Taurine Transporter] in Holstein-Friesian dairy cattle [69]. Presence of TauT gene in cattle leads to taurine deficiency and resulting immune system abnormality [70]. Moreover, higher productivity of dairy cows predisposes them to metabolic stress. Metabolic stress ultimately compromises the immune status of the animals and thereby their risk to be infected with different infectious diseases likes tuberculosis [71,72].

Coughing is an important clinical sign of respiratory disorders. Various factors are associated with animals cough. The current study found that herd-level factors such as poor floor sanitation (77.77%), cattle in same premises (92.59%), defective ventilation (94%) and improper waste disposal system (94.18%) which may be directly or indirectly associated with cough in cattle and might be increased the prevalence of bTb in cattle. Many authors stated that sanitary condition, poor husbandry, improper ventilation, close contact of animal are the important factors for the presence of cough which may increase the prevalence of bTb in cattle [23,27,49,72–74].

The median cattle herd size we observed was 4, which are also true for other districts in Bangladesh [23]. The cattle management system of the study districts and most other districts (except few like Sirajganj and coastal districts) is also similar. So, the true animal-level prevalence of bTb can be extrapolated to most other districts in Bangladesh. To know the true burden of bTb further studies including larger herds is recommended.

Our results indicate that crossbred, non-grazing cows with cough should be targeted in a prospective bTb surveillance program. Targeted sampling of cows with the aforementioned characteristics will increase the probability of detecting infected animals and herds. Herds in the regions of Mymensingh Sadar and Bhaluka experience a higher bTb true prevalence and should be a priority in future control programs. The Bayesian framework allowed for the structured, hierarchical true prevalence estimation that identified the extent of bTb presence at the region, herd and animal level. Thus, control efforts can be region and herd- specific depending on the true bTb prevalence of infection.

## Supporting information

**S1 File. OpenBUGS code to estimate hierarchical prevalence of bovine tuberculosis.**
(TXT)

**S2 File. Data used for the estimation of hierchical true prevalence of bovine tuberculosis in Bangladesh.**
(CSV)

**S3 File. Data used for the identification of herd level risk factors for bovine tuberculosis in Bangladesh.**
(CSV)

**S4 File. Data used for the identification of animal level risk factors for bovine tuberculosis in Bangladesh.**
(CSV)

**S5 File. Univariable association of independent variables with herd level bovine tuberculosis in Mymensingh district, Bangladesh.**
(DOCX)

**S6 File. Univariable association of independent variables with animal level bovine tuberculosis in Mymensingh district, Bangladesh.**
(DOCX)

## Acknowledgments

The authors are grateful to farmers for providing the samples and data for this study.

## Author Contributions

**Conceptualization:** Polychronis Kostoulas, A. K. M. Anisur Rahman, Md. Mahbub Alam.

**Data curation:** Md. Nazimul Islam, Mohammad Kamruzzaman Khan.

**Formal analysis:** Md. Nazimul Islam, Mohammad Kamruzzaman Khan, Mohammad Ferdousur Rahman Khan, Polychronis Kostoulas, A. K. M. Anisur Rahman.

**Funding acquisition:** Md. Mahbub Alam.

**Investigation:** A. K. M. Anisur Rahman, Md. Mahbub Alam.

**Methodology:** Md. Nazimul Islam, Mohammad Kamruzzaman Khan, Mohammad Ferdousur Rahman Khan, Polychronis Kostoulas, A. K. M. Anisur Rahman, Md. Mahbub Alam.

**Project administration:** Md. Mahbub Alam.

**Software:** Md. Nazimul Islam, Mohammad Kamruzzaman Khan, Polychronis Kostoulas, A. K. M. Anisur Rahman.

**Supervision:** Mohammad Ferdousur Rahman Khan, A. K. M. Anisur Rahman, Md. Mahbub Alam.

**Writing – original draft:** Md. Nazimul Islam, Mohammad Kamruzzaman Khan.

**Writing – review & editing:** Mohammad Ferdousur Rahman Khan, Polychronis Kostoulas, A. K. M. Anisur Rahman, Md. Mahbub Alam.

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
