## [Decision Letter · Decision Letter 0]

6 Jan 2021

PONE-D-20-31243

Bovine tuberculosis true prevalence and risk factors in Bangladesh

PLOS ONE

Dear Dr. Alam,

Thank you for submitting your manuscript to PLOS ONE. After careful consideration, we feel that it has merit but does not fully meet PLOS ONE’s publication criteria as it currently stands. Therefore, we invite you to submit a revised version of the manuscript that addresses the points raised during the review process.

We look forward to receiving your revised manuscript.

Kind regards,

Seyed Ehtesham Hasnain

Academic Editor

PLOS ONE

Journal Requirements:

2. During your revisions, please note that a simple title correction is required: to follow correct English language usage, the title should read "Risk factors and prevalence of bovine tuberculosis in Bangladesh". Please ensure this is updated in the manuscript file and the online submission information.

3.Thank you for stating the following financial disclosure:

 "The funders had no role in study design, data collection and analysis, decision to publish, or preparation of the manuscript"

5. We note you have included a table to which you do not refer in the text of your manuscript. Please ensure that you refer to Table 5 and 6 in your text; if accepted, production will need this reference to link the reader to the Table.

Additional Editor Comments:

Major Revision

Reviewers' comments:

Reviewer's Responses to Questions

**Comments to the Author**

1. Is the manuscript technically sound, and do the data support the conclusions?

Reviewer #1: Yes

Reviewer #2: Yes

2. Has the statistical analysis been performed appropriately and rigorously? 

Reviewer #1: Yes

Reviewer #2: I Don't Know

3. Have the authors made all data underlying the findings in their manuscript fully available?

Reviewer #1: Yes

Reviewer #2: Yes

4. Is the manuscript presented in an intelligible fashion and written in standard English?

Reviewer #1: Yes

Reviewer #2: Yes

5. Review Comments to the Author

Reviewer #1: Nazimul Islam et al in their study entitled "Bovine tuberculosis true prevalence and risk factors in Bangladesh" have discussed several epidemiological parameters and risk factor which are associated with the bovine tuberculosis in their region

I have following observation / concerns which need attention by the author before this paper can be considered for the publication

1. Whether author examined sero prevelance of M. paratuberculosis in their cohort.

2. It would be advantageous if blood hematogram and immune cells analysis is done to corelate with the potential reason of sensitivity.

3. Adding some description of immune genetic aspect ( MHC gene e.g. ) would also help in understanding the reason of sensitivty of the animals for BTB

Reviewer #2: The authors in the manuscript planned to determine the true prevalence and identify risk factors for bTB at the animal- and herd-level in Bangladesh. Considering the endemic nature and economic consequences of bovine TB, this work has potential to determine the true prevalence so that effective interventional strategies can be put in place. A major issue is that there has been a recent related publication from same university as mentioned below, though not referred in the manuscript. Authors need to be wary of the overlapping results and observations.

Islam SKS, Rumi TB, Kabir SML, van der Zanden AGM, Kapur V, Rahman AKMA, et al. (2020) Bovine tuberculosis prevalence and risk factors in selected districts of Bangladesh. PLoS ONE 15(11): e0241717. https://doi.org/10.1371/journal.pone.0241717

There are some minor changes that need to be incorporated as mentioned below

The Affiliation no. 5 has not been assigned to any author.

Please rephrase line 53.

There is large scale discrepancy in CFT and CCT results that may be explained in discussion.

Result headings need to be more descriptive to include result summary.

Characteristics of the test samples may be provided like gender, age and physical conditions etc.

6. PLOS authors have the option to publish the peer review history of their article (what does this mean?). If published, this will include your full peer review and any attached files.

Reviewer #1: No

Reviewer #2: **Yes: **Javaid Ahmad Sheikh

---

## [Author Response · Author response to Decision Letter 0]

3 Feb 2021

We thank both of the reviewers and editor for their valuable comments which has led to an improved manuscript. The changes we made have been shown in blue font.

Reviewer #1: Nazimul Islam et al in their study entitled "Bovine tuberculosis true prevalence and risk factors in Bangladesh" have discussed several epidemiological parameters and risk factor which are associated with the bovine tuberculosis in their region

I have following observation / concerns which need attention by the author before this paper can be considered for the publication

1. Whether author examined sero prevelance of M. paratuberculosis in their cohort.

Response: We did not examine seroprevalence of M. paratuberculosis in the skin test positive animals. It could provide information about the cross-reaction of M. paratuberculosis among skin test positive animals. As per OIE guideline, initially we screened all selected animals by caudal fold tuberculin test [CFT] (test positive results may include responses due to M. paratuberculosis also) using bovine purified protein derivatives [PPD] then CFT positive animals were further confirmed by comparative cervical tuberculin test [using both avian and bovine PPDs] which excluded responses from M. paratuberculosis. Moreover, we used serial interpretation of the diagnostic tests which decreases false positive results and hence increases the positive predictive value of the diagnostic tests. We have discussed this in the revised manuscript. Lines: 273-281.

2. It would be advantageous if blood hematogram and immune cells analysis is done to corelate with the potential reason of sensitivity.

Response: Blood hematogram and immune cells analysis were not performed. Immune cell analysis is costly and requires sophisticated laboratory facilities which lack in resource limited settings like Bangladesh. Skin tests are routinely used in Bangladesh for the diagnosis of bovine tuberculosis which is an OIE recommended test for international trade also. 

3. Adding some description of immune genetic aspect (MHC gene e.g. ) would also help in understanding the reason of sensitivty of the animals for BTB.

Response: We have added some description of the immunogenetic aspectic of bTB susceptibility in the revised manuscript. Lines: 300-305.

Reviewer #2: The authors in the manuscript planned to determine the true prevalence and identify risk factors for bTB at the animal- and herd-level in Bangladesh. Considering the endemic nature and economic consequences of bovine TB, this work has potential to determine the true prevalence so that effective interventional strategies can be put in place. A major issue is that there has been a recent related publication from same university as mentioned below, though not referred in the manuscript. Authors need to be wary of the overlapping results and observations.

Islam SKS, Rumi TB, Kabir SML, van der Zanden AGM, Kapur V, Rahman AKMA, et al. (2020) Bovine tuberculosis prevalence and risk factors in selected districts of Bangladesh. PLoS ONE 15(11): e0241717. https://doi.org/10.1371/journal.pone.0241717

Response: During submission of this manuscript the above mentioned manuscript was not published and hence not cited. Now, in the revised manuscript we have checked for any overlapping results and observation and cited [47]. Lines 271-273 and lines 289-290.

There are some minor changes that need to be incorporated as mentioned below

The Affiliation no. 5 has not been assigned to any author.

Response: The affiliation no. 5 has now been assigned to one author. 

Please rephrase line 53.

Response: Line 53 rephrased as suggested by the reviewer. 

There is large scale discrepancy in CFT and CCT results that may be explained in discussion.

Response: We have now some description of the discrepancy in CFT and CCT results in the revised manuscript. Lines: 263-267.

Result headings need to be more descriptive to include result summary.

Response: I have added some subheadings in the result section. 

Characteristics of the test samples may be provided like gender, age and physical conditions etc.

Response: We provided those in the submitted manuscript as two supplementary files [supplementary files 2 and 3]

---

## [Editor Report · Decision Letter 1]

15 Feb 2021

Risk factors and true prevalence of bovine tuberculosis in Bangladesh

PONE-D-20-31243R1

Dear Dr. Alam,

We’re pleased to inform you that your manuscript has been judged scientifically suitable for publication and will be formally accepted for publication once it meets all outstanding technical requirements.

Kind regards,

Seyed Ehtesham Hasnain

Academic Editor

PLOS ONE

Additional Editor Comments (optional):

The manuscript was sent for revision and Authors have modified the manuscript keeping in mind the comments of the Reviewers. I have gone through the revised manuscript and also the Authors response to the comments of the reviewers. Authors have added some description of immunogenetic aspectic of bTB susceptibility in the revised manuscript. In the revised manuscript, Authors have also added some description related to the discrepancy in the CFT and CCT results in the discussion part. In my view, the authors have satisfactorily addressed all the comments made by the reviewers and added all required information, and have revised the manuscript accordingly. I recommend this manuscript for publication.
---

## [Editor Report · Acceptance letter]

19 Feb 2021

PONE-D-20-31243R1 

Risk factors and true prevalence of bovine tuberculosis in Bangladesh 

Dear Dr. Alam:

I'm pleased to inform you that your manuscript has been deemed suitable for publication in PLOS ONE. Congratulations! Your manuscript is now with our production department. 

Kind regards, 

on behalf of

Prof Seyed Ehtesham Hasnain 

Academic Editor

PLOS ONE